

# Recent variations in oceanic transports across the Greenland-Scotland-Ridge

Michael Mayer[1,2,3], Takamasa Tsubouchi[4], Susanna Winkelbauer[1], Karin Margretha H. Larsen[5], Barbara Berx[6], Andreas Macrander[7], Doroteaciro Iovino[8], Steingrímur Jónsson[7,9], Richard Renshaw[10]

[1]Department of Meteorology and Geophysics, University of Vienna, Vienna, 1090, Austria

[2]Research Department, European Centre for Medium-Range Weather Forecasts (ECMWF), Bonn, 53175, Germany

[3]b.geos, Korneuburg, 2100, Austria

[4]Atmosphere and Ocean Department, Japan Meteorological Agency (JMA), Tokyo, 105-8431, Japan

[5]Faroe Marine Research Institute, Tórshavn, FO-100, Faroe Islands

[6]Marine Scotland, Aberdeen, AB11 9DB, United Kingdom

[7]Marine and Freshwater Research Institute, Hafnarfjörður, 220, Iceland

[8]Ocean Modeling and Data Assimilation Division, Centro Euro-Mediterraneo sui Cambiamenti Climatici (CMCC), Bologna, 40127, Italy

[9]University of Akureyri, Akureyri, 600, Iceland

[10]Met Office, Exeter, EX1 3PB, United Kingdom

*Correspondence to*: Michael Mayer (Michael.mayer@univie.ac.at)






**Abstract.** Oceanic exchanges across the Greenland-Scotland-Ridge (GSR) play a crucial role in shaping the Arctic climate and linking with the Atlantic Meridional Overturning Circulation. The considered ocean reanalyses underestimate the observed 1993-2021 mean inflow of warm and saline Atlantic Waters of $8.0 \pm 0.5$ Sv by 7-23%, with reanalyses at 1/4° resolution exhibiting larger biases than higher resolution products. The underestimation of Atlantic Water inflow translates

into a low bias in mean oceanic heat flux at the GSR of 4-32% in reanalyses compared to the observed value of $281 \pm 18$ TW. Interannual variations of reanalysis transports correlate reasonably well with observed transports in most branches crossing the GSR. Observations and reanalyses with data assimilation show a marked reduction of oceanic heat flux across the GSR of 4-10% (compared to 1993-2021 means) during a biennial (two-year-long) period centered on 2018, a record-low for several products. The anomaly was associated with a temporary reduction of geostrophic Atlantic Water inflow through

the Faroe-Shetland Branch and was augmented by anomalously cool temperatures of Atlantic waters arriving at the GSR. The latter is linked to a recent strengthening of the North Atlantic Subpolar Gyre and illustrates the interplay of interannual and decadal changes in modulating transports at the GSR.

**Short Summary.** This paper compares oceanic fluxes across the Greenland-Scotland-Ridge (GSR) from ocean reanalyses to

largely independent observational estimates. All reanalyses underestimate the inflow of warm waters of subtropical Atlantic origin and hence oceanic heat transport across the GSR. Investigation of a strong negative heat transport anomaly around 2018 highlights the interplay of variability on different time scales and the need for long-term monitoring of the GSR to detect forced climate signals.




## 1 Introduction

The Greenland-Scotland-Ridge (GSR), encompassing Denmark Strait, Iceland-Faroe (I-F) Ridge, Faroe-Shetland (F-S) Channel, and the European shelf, represents the main oceanic gateway to the Arctic Mediterranean (the ocean bounded by the GSR, Davis Strait, and Bering Strait). Oceanic transports across the GSR play an important role for the Arctic and global climate systems. In the surface layer, the warm and saline Atlantic Water moves northward across the GSR and the light Polar Water flows southward mainly through the Denmark Strait. In the lower layer, the cold and dense water is transported

southward at depth into the North Atlantic, contributing to the lower limb of the Atlantic Meridional Overturning Circulation (Hansen and Østerhus 2000; Buckley and Marshall 2016).

Transports across the GSR exhibit pronounced interannual variability, and thereby play an important role in modulating the heat budget of the Arctic Mediterranean (e.g., Muilwijk et al. 2018, Mayer et al. 2016, or Asbjørnsen et al. 2019). Specifically, the inflow of warm and saline Atlantic Water (AW) exhibits a strong co-variability with ocean heat content,

especially of the AW layer of the Arctic Mediterranean (Mayer et al. 2022). Tsubouchi et al. (2021), using observation-based oceanic transport data 1993-2016, revealed a step-change towards stronger oceanic heat transports (OHTs) across the GSR around 2002, suggesting an enhanced contribution of OHT to the observed warming of the Arctic Ocean. Mayer et al. (2022) temporally extended the monitoring of OHT at the GSR using ocean reanalyses and found a pronounced reduction of OHT around 2018, which could not be verified with observational data at that time and the causes of which were not explored in

detail.

Here, we use observational oceanic transport data at the boundaries of the Arctic Mediterranean updated to 2021 and an extended set of ocean reanalyses to explore the pronounced reduction of OHT in more detail, track it to the main contributing oceanic branch, and relate these changes to larger-scale climate variations on interannual and decadal time scales. An additional aspect of this study is a more detailed validation of reanalysis-based oceanic transports at the GSR at

the scale of single branches, to further build trust in the usefulness of these products for monitoring Arctic climate and its oceanic drivers.

## 2 Data and methods

We use monthly data from a comprehensive set of ocean reanalyses to compute oceanic transports across GSR, Davis Strait,

and Bering Strait. The latter two straits are calculated to close the volume budget and obtain unambiguous net heat transport into the Arctic Mediterranean (Schauer et al. 2009). Inflow (positive) has been defined as positive towards the Arctic Mediterranean. The employed products are an updated ensemble based on the Copernicus Marine Environment Monitoring Service Global Reanalysis Ensemble Product (CMEMS GREPv2, product ref 1), consisting of ORAS5, CGLORS,



GLORYS2V4, and GloRanV14 [an improvement of GloSea5 (MacLachlan et al., 2015), also known as the FOAM product; product ref 6]. These are all run at 1/4° horizontal resolution with 75 vertical levels and use atmospheric forcing from ERA-Interim (Dee et al. 2011). The ensemble is complemented with GLORYS12V1 (product ref 2), a reanalysis at 1/12° horizontal resolution with 50 vertical levels driven by ERA-Interim atmospheric forcing, and GLOB16 (product ref 5), a hindcast ocean simulation (i.e. with no data assimilation) at 1/16° horizontal resolution and 98 vertical levels (Iovino et al., 2016) driven by JRA55-do (Tsujino et al., 2018). Transports are computed on the native grid through line integrals similar to Pietschnig et al. (2018).

Observational mass-consistent estimates of oceanic transports (product ref 3) are updated to July 2021 following Tsubouchi et al. (2021; i.e., using the same strategy to infill data gaps, uncertainty estimation, and box inverse model to close the volume budget). Temporal coverage and references for the single observational estimates used as input are provided in the data table (Table 1). Surface freshwater inputs by river discharge and precipitation minus evaporation for 1993-2021 as input to the box inverse model are based on Winkelbauer at al. (2022). The used reanalyses assimilate temperature and salinity profiles available from data bases such as Hadley EN4 (Good et al. 2013), which according to our investigations include only a small subset of the mooring data used for our observational transport estimates. Currents are generally not assimilated in ocean reanalyses. Hence, the observation-based volume fluxes represent fully and temperature fluxes largely independent validation data.

As in Mayer et al. (2022), we assume total uncertainties of monthly mean observations (provided in Tsubouchi et. al 2021) to consist of roughly half systematic and half random errors, i.e. the two contributions are the total uncertainty reduced by a factor of $\frac{1}{\sqrt{2}}$, respectively. Consequently, the contribution of random errors to uncertainties of long-term mean observational estimates are further reduced by a factor of $\frac{1}{\sqrt{N}}$, where N is the number of years, and deseasonalized anomalies only include the random errors.

Transported water masses at the GSR are decomposed into Atlantic (AW), Polar (PW), and Overflow Waters (OW), largely following Eldevik et al. (2009). PW is defined as T<4°C and $\sigma_\theta$ <27.7kg/m3. OW is defined as $\sigma_\theta$ > 27.8kg/m3. The rest (i.e., waters with $\sigma_\theta$<27.8kg/m3 with PW taken out) is considered AW. Note, these definitions have been revised from Mayer et al. (2022). Water mass decomposition is performed each month based on the monthly T and S fields in the reanalyses. These definitions are similar to those used for observational products (see references for more details).

We additionally use sea level anomaly (SLA) data provided through CMEMS (product ref 4) for investigating drivers of observed OHT anomalies. The global mean SLA trend is removed before computation of the presented diagnostics.

Deseasonalized anomalies are based on the 1993-2019 climatologies, i.e. the period for which all data are available at the moment. Anomaly time series have a 12-monthly temporal smoother applied to emphasize interannual variations.





Confidence levels (95% is set as threshold for significance testing) for temporal correlations take auto-correlation of the

involved time series into account (see Oort and Yienger 1996).

## 3 Results

### 3.1 General evaluation of transports of water masses across GSR

Figure 1 presents mean annual cycles and anomaly time series of relevant oceanic transport quantities at the GSR. It is
complemented with long-term averages shown in Table 2. Observations show seasonally varying AW inflow across the GSR
($8.0\pm0.5$ Sv; mean $\pm$ standard deviation of the mean reported throughout unless explicitly stated) with a maximum in
December and minimum in June-July (Fig. 1a). The AW inflow is largely balanced by PW (Fig. 1c; $-1.7\pm1.0$ Sv) and OW
(Fig. 1b; $-5.6\pm0.3$ Sv) outflow, yielding a relatively small net volume flux across the GSR of 0.7 Sv (balanced by flows
through Bering and Davis Straits). The PW outflow exhibits an annual cycle balancing the AW inflow (i.e., maximum
outflow in boreal winter), while the OW exchange is more stable throughout the year (i.e., small annual cycle).

All reanalyses underestimate AW inflow across the GSR when compared to observations, but the shape of the annual cycle
of all estimates is in good agreement with observations. The 1/4° products - except for CGLORS - tend to have a larger low
bias than the high-resolution products. Agreement is also good for the PW outflow, where all products show the observed
seasonal maximum flow in boreal winter. The range of reanalysis-based estimates is large in a relative sense, with the
observations lying in the middle of the range. There is less coherence across products concerning the OW transports.
GLORYS12, GLORYS2V4, and CGLORS are close to observations, with rather persistent overflow on the order of -5.5 Sv,
with seasonal variations that agree with the observations. Other products tend to have overflows that are too weak (most
notably ORAS5), and some also exhibit biases in the representation of the annual cycle (e.g., GloRanV14).

Table 2 additionally includes long-term average volume fluxes in the main AW inflow branches (North Icelandic Irminger
Current (NIIC), I-F branch, F-S branch, and the European Shelf). The reanalysis-based estimates agree generally well with
observations. The main discrepancy is the underestimation of I-F inflow and overestimation of F-S inflow by all GREP
reanalyses, while the high-resolution products GLORYS12 and GLOB16 are in much better agreement with observations.
This suggests that increased resolution, along with more realistic bathymetry, improves representation of inflow pathways in
the reanalyses. We also note that temporal anomaly correlations with observed I-F volume fluxes are very low (Pearson
correlation coefficients r range in -0.07 to 0.25 and are statistically insignificant) for all reanalyses, but are substantially
higher for F-S volume fluxes (r ranges in 0.31 to 0.74, statistically significant for all products with data assimilation).





Fig 1d shows the mean annual cycle of heat flux across the GSR, i.e. the sum of sensible heat transported by all waters crossing the GSR. The mean annual cycle of GSR heat fluxes generally follows that of AW volume fluxes, with a minimum between boreal spring/early summer and a maximum in fall/early winter. Seasonal minima and maxima in GSR heat flux co-occur with those of AW volume flux, i.e. seasonal variations in heat flux are largely volume-flux-driven and the seasonal cycle in volume-weighted temperatures is in phase.

Since net volume flux across the GSR is small, the ambiguity arising from the choice of reference temperature can be considered small as well. However, for the long-term averages we focus on net heat transport into the Arctic Mediterranean, i.e. the sum of heat fluxes across the GSR, plus those through Bering and Davis Straits. Values in Table 2 show that all reanalyses exhibit lower net heat transport (by ~24% for the GREP mean) than that observed (306±19 TW; 311±20 TW when including sea ice), with the high-resolution products performing clearly better than the GREP (i.e., underestimation reduced to ~15%). We note however that GLOB16 exhibits a strong warm bias (of order 1.5°C) in AW, i.e. the relatively high heat transport seems to be achieved for a wrong reason. Based on Mayer et al. (2022) and taking account of oceanic storage, the energy-budget-based estimate of the net heat boundary transport suggests even higher values (~348 TW) than observations. Table 2 also confirms that long-term averages for heat flux across the GSR are qualitatively very similar to the net heat transport, i.e., heat fluxes across the GSR are the dominant contributor to oceanic heat transport into the Arctic Mediterranean.

Fig 1e shows deseasonalized anomalies of AW volume flux, with a 12-monthly smoother applied to emphasize interannual variability. Typical variability is similar across observations (temporal standard deviation $\sigma = 0.35$ Sv) and reanalyses ($\sigma$ ranges in 0.22 to 0.47 Sv). Temporal correlations between reanalysed and observed AW inflow anomalies are moderately high (r ranges in 0.42 and 0.62, see legend of the plot for values, and is statistically significant for all products).

Fig 1f shows anomalies of total oceanic heat flux across the GSR, which show similar variability as AW volume flux, i.e. the strength of AW inflow not only modulates the seasonal cycle of the total GSR heat flux, but also its interannual variations (r ranges in 0.86 to 0.91). GSR total heat flux from reanalyses is in slightly better agreement with observations than AW volume fluxes (r ranges in 0.48 and 0.64, see legend of plot for values, and is statistically significant for all products). We note that GLOB16 exhibits the weakest correlation with observed heat flux, which is likely related to the fact that this product does not assimilate any ocean parameters (e.g., surface or subsurface temperatures, sea surface height). Fig 1f  also shows a prominent negative heat flux anomaly centered around the year 2018, which has already been noted by Mayer et al. (2022) for net heat transport into the Arctic Mediterranean.

160

**3.2 Spatial structure of the 2017/07-2019/06 transport anomaly**





To set the scene for further investigation, we present climatological temperatures and currents at the GSR based on GLORYS12 in Fig. 2a and b, respectively. Comparison with analogous figures based on the GREP [shown in Mayer et al. (2022)] allows to appreciate the benefits of increased resolution (1/12° vs 1/4° resolution), including a more distinct representation of inflow and outflow branches and a spatially more variable bathymetry, especially in the I-F branch.

Next, we investigate in more detail the recent reduction of AW volume and GSR heat fluxes. This is most prominent in the biennial (two-year-long) signal of average anomalies in 2017/07-2019/06. Fig. 2c shows that, during this period, strong warm anomalies were present over 0-400m depth in eastern Denmark Strait and warm anomalies are also seen in the F-S branch. The latter suggests a temporary deepening of the AW layer. Velocity anomalies for the 2017/07-2019/06 period (Fig. 2d) suggest that the positive temperature anomalies in eastern Denmark Strait are driven by enhanced NIIC transports. The strongest and deepest velocity anomaly in 2017/07-2019/06 is located in the eastern part of the F-S branch, where reduced inflow is present from the surface down to the interface at ~600m, and hence we focus on this branch next. Negative anomalies are also seen in the overflow from 600m to 1000m depth. There is some compensation by positive velocity anomalies in western F-S (meaning reduced southward flow there), but the effect of the eastern F-S anomaly dominates, and the net F-S volume flux anomaly was clearly reduced during this period (see below). We note that these main features are similar also in anomaly sections based on the GREP ensemble mean (not shown).

Observations and all reanalyses show large negative F-S volume inflow anomalies of -0.62Sv/-0.60Sv/-0.33Sv/-0.20Sv in observations/GREP_mean/GLORYS12/GLOB16, respectively, during 2017/07-2019/06 (Fig. 2e). In four out of seven datasets, this is the overall biennial minimum of the 1993-2021 record (not shown). The total AW volume flux anomaly for that period was -0.26Sv/-0.44Sv/-0.48Sv/+0.17Sv in observations/GREP_mean/GLORYS12/GLOB16, respectively. Thus, F-S volume flux anomalies were partly compensated by other AW branches, with no very clear signal in any of them (not shown). Only GLOB16 exhibits a positive AW volume flux anomaly during that period, which appears to be related to a shift towards generally higher AW volume flux around 2016 (see Fig. 1e). This is not seen in any of the other products.

Temperature transport anomalies in the F-S branch are strongly correlated with volume flux anomalies (compare Fig. 2e and f), and there is a clear reduction of F-S branch temperature flux in 2017/07-2019/06 by -22.1TW/-21.8TW/-15.4TW/-7.7TW in observations/GREP_mean/GLORYS12/GLOB16. The contribution of temperature anomalies in the F-S Channel during that time was small, with biennial anomalies of observed volume-weighted temperatures between -0.2 and -0.1 K (similar for all products, except GLOB16 with a positive anomaly). The total AW heat flux anomaly for 2017/07-2019/06 was -11.9TW/-19.5TW/-21.6TW/+7.4TW in observations/GREP_mean/GLORYS12/GLOB16. The positive value of GLOB16 being related to its potentially spurious positive AW volume flux anomaly. Thus, the reduction of GSR heat transports during that period was mainly driven by a reduction of volume inflow through the F-S branch. It was only partly offset by compensating transport anomalies in other branches. Very similar biennial heat transport anomalies for the total GSR (-





11.9TW/-18.8TW/-22.8TW/+0.2TW for observations/GREP_mean/GLORYS12/GLOB16) confirm AW as the main driver of heat flux variability across the GSR.


## 3.3 Relationships between sea level and AW inflow

For a better understanding of mechanisms contributing to the GSR transport anomaly around 2018, we first consider simple statistical relationships between SLA and oceanic quantities. We find a statistically significant temporal correlation of the zonal SLA gradient at GSR with observed AW volume flux anomalies (Fig. 3a), which is plausible in terms of geostrophic

balance. The correlation pattern looks very similar when performed with the F-S branch volume flux alone (not shown). The pattern of temporal correlation between the SLA field and observed anomalies of volume-weighted temperature of AW transports (Fig. 3b) is distinct from the relationship with AW volume flux (compare Fig. 3a). This emphasizes SLA in the North Atlantic Subpolar Gyre (SPG), with higher SLA in the SPG (i.e., a weaker gyre) associated with higher volume-weighted temperature, and vice versa. Although correlations are high (r up to 0.69) in the SPG region, they are not

statistically significant. The cause may be the low number of degrees of freedom, as the SLA in the SPG exhibits high temporal auto-correlation (see Fig. 3d discussed below).

Actual SLA anomalies averaged over 2017/07-2019/06 (Fig. 3c) indicate a weakened zonal SLA gradient at the GSR, albeit not very strongly pronounced, and anomalously low SLA in the SPG region. According to the correlation patterns discussed above, these two features suggest reduced AW volume flux (as suggested by patterns in Fig. 3a) and anomalously low AW

volume-weighted temperatures (as suggested by patterns in Fig. 3b). We also note the positive SLA anomalies north of the GSR with a maximum in the central Nordic Seas.

To put these results in context, we define two SLA-based indices (Fig. 3d) from the correlation patterns found in Figs. 3a and b. The similarity of correlations in Fig. 3a to the correlation between total OHT at the GSR and SLA shown in Mayer et al. (2022) reinforces use of their gradient-based index (i.e., standardized SLA difference between 58-60 N / 2-0 W and 63-67 N

/ 20-15 W). This index is correlated with AW volume flux anomalies (r=0.62 for observed transports and ranges in 0.47 to 0.70 for reanalyses - statistically significant in all cases). The second index uses spatial SLA averages in the North Atlantic region (55 - 60 N and 40 - 15 W) as an inverse proxy of SPG strength. This index is correlated with anomalies of volume-weighted temperatures of AW transports. The two indices in Fig. 3d show different characteristics, with the SPG index varying on decadal time scales, while the gradient index shows stronger interannual variations. The SPG index has been

negative after ~2014, which suggests a strong SPG and lower inflow temperatures in recent years, in agreement with results by Hátún and Chafik (2018). This SPG index exhibits two minima during the 2017/07-2019/06 period, although not extreme relative to the entire time series.





Fig. 3e shows volume-weighted temperatures of AW waters from different products and confirms their overall decrease in recent years. Comparison with the SPG index in Fig. 3d suggests a generally delayed response of volume-weighted

temperature in Atlantic inflow water at the GSR to SPG strength. No robust statements can be made about the evolution of volume-weighted temperatures in 2020/21, as insufficient datasets were updated at the time of writing (see data statement in section 4).

## 4 Data availability

The data products used in this article, as well as their names, availabilities and documentations are summarized in Table 2.

**Note to reviewers: Use of data up to 2021 is a requirement for the Ocean State Report 7. However, not all employed products have been updated to 2021 by the data providers by the time of submission. This becomes apparent from some time series becoming flat in 2020 and/or 2021. We plan to update the data during the revision phase. This is not expected to change the outcomes of this study.**

## 235  5 Conclusions

Reanalysis-based oceanic transports show generally good agreement with observations on the scale of single branches of the GSR, both in terms of mean and variability of volume and heat fluxes. There is some indication that the higher resolution products have a better representation of AW inflow in the I-F and F-S branches. The higher-resolution products also tend to have a higher net heat flux into the Arctic Mediterranean, which brings them closer to observations. The energy-budget-

based estimate from Mayer et al. (2022) suggests even higher net heat flux than oceanic observations, which confirms the underestimation of transports by the ocean reanalyses and indicates that observations possibly miss some influx of heat. This apparent underestimation was not found in an analogous comparison of oceanic transports into the central Arctic and an energy-budget-based estimate (Mayer et al. 2019), potentially reflecting the different observational strategies in Fram Strait and Barents Sea Opening compared to the GSR (see, e.g., Dickson et al. 2008).

All reanalyses with data assimilation and observations show a pronounced reduction of OHT during the two-year period 2017/07-2019/06, with some recovery after that. Comparison of observed SLA patterns during this period with statistical relationships between SLA and oceanic transports suggests that this reduction arose from a combination of interannual (i.e., reduced zonal SLA gradient at the GSR) and decadal scale changes (i.e., strong SPG in recent years). Another potential factor contributing to the OHT reduction during 2017/07-2019/06 may have been the positive SLA anomalies centred in the

Nordic Seas (Fig 3c), which Chatterjee et al. (2018) have related to a weakened gyre circulation in the Nordic Seas and may have contributed to the weakened AW inflow as well.





Our results also reveal a delayed response of AW inflow temperatures to SPG strength. This is consistent with earlier studies finding anti-correlation between SPG strength and GSR heat transport (Häkkinen et al., 2011; Hátún et al., 2005). Specifically, the generally weaker SPG during ~1997 and ~2014 (with more pronounced minima in 1997-98, 2010-11, and 2003-06, see Hátún and Chafik, 2018) was associated with warm inflow temperatures and stronger OHT after 2001 (Tsubouchi et al. 2021). After that, the SPG strengthened and the inflow temperatures declined, which is also consistent with generally reduced oceanic heat transports in recent years.

Our results indicate that decadal predictions of the SPG strength, which have been shown to exhibit skill (e.g., Robson et al. 2018 and Borchert et al. 2021), may also allow to infer near-term trends in OHT across the GSR. Another implication is that the strong interannual-to-decadal variability of OHT across the GSR hampers detection of longer-term (forced and unforced) trends in observed OHT, an aspect in which climate simulations show large spread (Burgard and Notz 2017). Continued in-situ monitoring of OHT, complemented with reanalysis efforts, is thus needed to provide observational time series of sufficient length for climate model validation.

**Acknowledgments**

M. Mayer and S. Winkelbauer received funding from Austrian Science Fund project P33177 and CMEMS 21003-COP-GLORAN Lot 7. The Iceland-Faroe branch and Faroe Bank Channel overflow data-collections received funding from the Danish Ministry of Climate, Energy and Utilities through its climate support program to the Arctic. The Denmark Strait Overflow time series was generated by Institution of Oceanography Hamburg and the Marine and Freshwater Research Institute (Iceland). They were supported through funding from Nordic WOCE, VEINS, MOEN, ASOF-W, NAClim, RACE II, RACE-Synthese, THOR, AtlantOS, and Blue Action (EU Horizon 2020 grant agreement nr. 727852).

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



**Tables**

| Ref. No. | Product name & type | Documentation |
|---|---|---|
| 1 | GLOBAL_REANALYSIS_PHY_001_031 (GREPv2) | PUM: http://marine.copernicus.eu/documents/PUM/CMEMS-GLO-PUM-001-031.pdf QUID: http://marine.copernicus.eu/documents/QUID/CMEMS-GLO-QUID-001-031.pdf https://doi.org/10.48670/moi-00024 |
| 2 | GLOBAL_MULTIYEAR_PHY_001_030 (GLORYS12V1) | PUM: https://catalogue.marine.copernicus.eu/documents/PUM/CMEMS-GLO-PUM-001-030.pdf QUID: https://catalogue.marine.copernicus.eu/documents/QUID/CMEMS-GLO-QUID-001-030.pdf |
| 3 | Mooring-derived ocean heat transport into Arctic Mediterranean from 1993 updated to July 2021 Observational input data (available from http://www.oceansites.org/tma/gsr.html): Iceland-Faroe branch: January 1993 to December 2020 Faroe-Shetland branch: January 1993 to June 2021 | Tsubouchi et al. (2021) Hansen et al. (2015) Berx et al. (2013) |





| | | |
|---|---|---|
| | North Icelandic Irminger Current: October 1994 to July 2021 | Jónsson and Valdimarsson (2012) |
| | Faroe Bank Channel: December 1995 to April 2021 | Hansen et al. (2016) |
| | Denmark Strait: May 1996 to 2021 available at https://www.cen.uni-hamburg.de/en/icdc/data/ocean/denmark-strait-overflow.html | Jochumsen et al. (2017) |
| | Bering Strait: August 1997 to August 2019 available at http://psc.apl.washington.edu/HLD/Bstrait/bstrait.html | Woodgate et al. (2018) |
| 4 | SEALEVEL_GLO_PHY_L4_REP_OBSERVATIONS_008_047 (DUACS) | QUID> https://catalogue.marine.copernicus.eu/documents/QUID/CMEMS-SL-QUID-008-032-062.pdf PUM: https://catalogue.marine.copernicus.eu/documents/PUM/CMEMS-SL-PUM-008-032-062.pdf |
| 5 | hindcast ocean simulation (no data assimilation) at 1/16° horizontal resolution and 98 vertical levels provided by CMCC (GLOB16) | Iovino et al. (2016) |
| 6 | ocean reanalysis at 1/4° horizontal resolution and 75 vertical levels provided by UKMO (GloRanV14) | MacLachlan et al. (2015) |

**Table 1. CMEMS and non-CMEMS products used in this study, including information on data documentation.**



### Volume flux 1993-2021 averages [Sv]

| | GSR Total | PW | OW | AW | AW NIIC | AW IF | AW FS | shelf |
|---|---|---|---|---|---|---|---|---|
| GREP (4 reanalyses) | 1.2 ± 0.5 | -1.3 ± 0.6 | -5.1 ± 0.4 | 6.8 ± 0.6 | 1.0 ± 0.1 | 1.9 ± 0.3 | 3.4 ± 0.6 | 0.5 ± 0.1 |
| GLORYS12 | 1.1 | -0.7 | -5.3 | 7.5 | 1.1 | 3.5 | 2.3 | 0.6 |
| GLOB16 | -0.2 | -2.6 | -4.7 | 7.1 | 0.6 | 3.4 | 2.6 | 0.5 |
| OBS | 0.7 ± 0.9 | -1.7 ± 1.0 | -5.6 ± 0.3 | 8.0 ± 0.5 | 0.9 ± 0.1 | 3.8 ± 0.3 | 2.7 ± 0.3 | 0.6 ± 0.3 |

### Temperature flux 1993-2021 averages [TW]

| | GSR Total | Arctic Mediterranean |
|---|---|---|
| GREP (4 reanalyses) | 219 ± 23 | 232 ± 21 |
| GLORYS12 | 248 | 258 |
| GLOB16 | 268 | 288 |
| OBS | 281 ± 18 | 306 ± 19 |
| Energy-budget based | - | 348 |

**Table 2. Long-term mean of (upper panel) volume flux in different water masses and branches across GSR and (lower panel) total heat transport across GSR and the Arctic Mediterranean. Observational European Shelf volume fluxes are based on Østerhus et al. (2019). The energy-budget-based transport estimate is taken from Mayer et al. (2022).**



# Figures

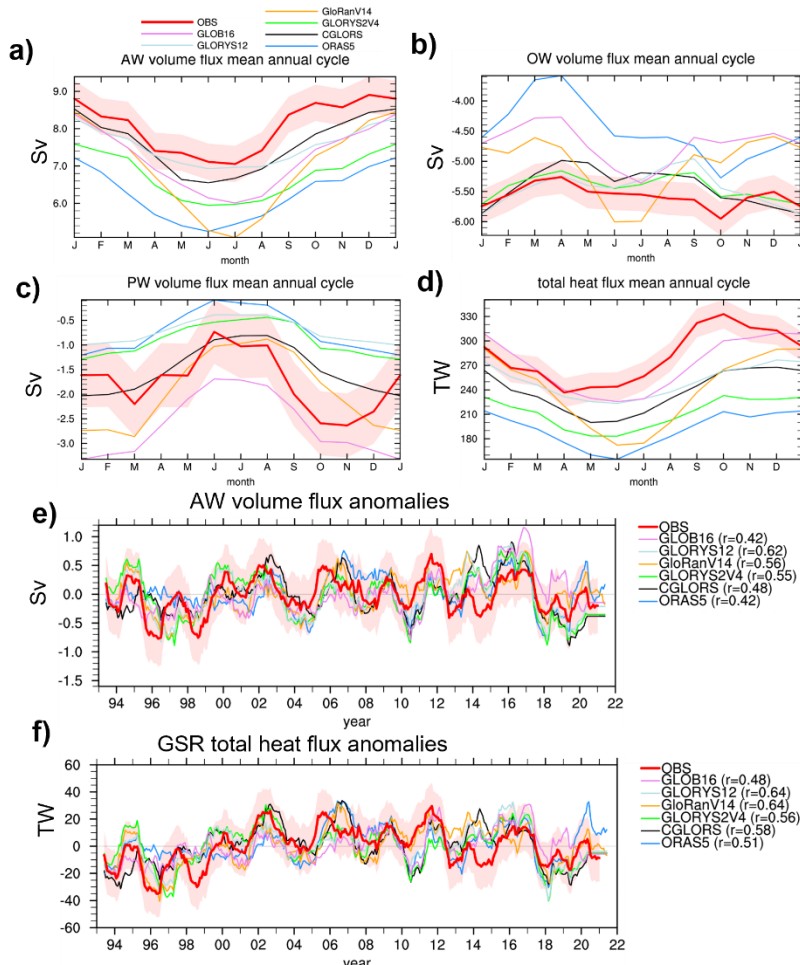

**Fig. 1. Mean annual cycle of (a) AW volume flux (b), OW volume flux, (c) PW volume flux, and (d) GSR total (AW+PW+OW) heat flux; Temporal anomalies of (e) AW volume flux and (f) GSR total heat flux. The red shading indicates ± 1 standard error of the observational data. Temporal correlations of reanalyses with observations are provided in the legends.**




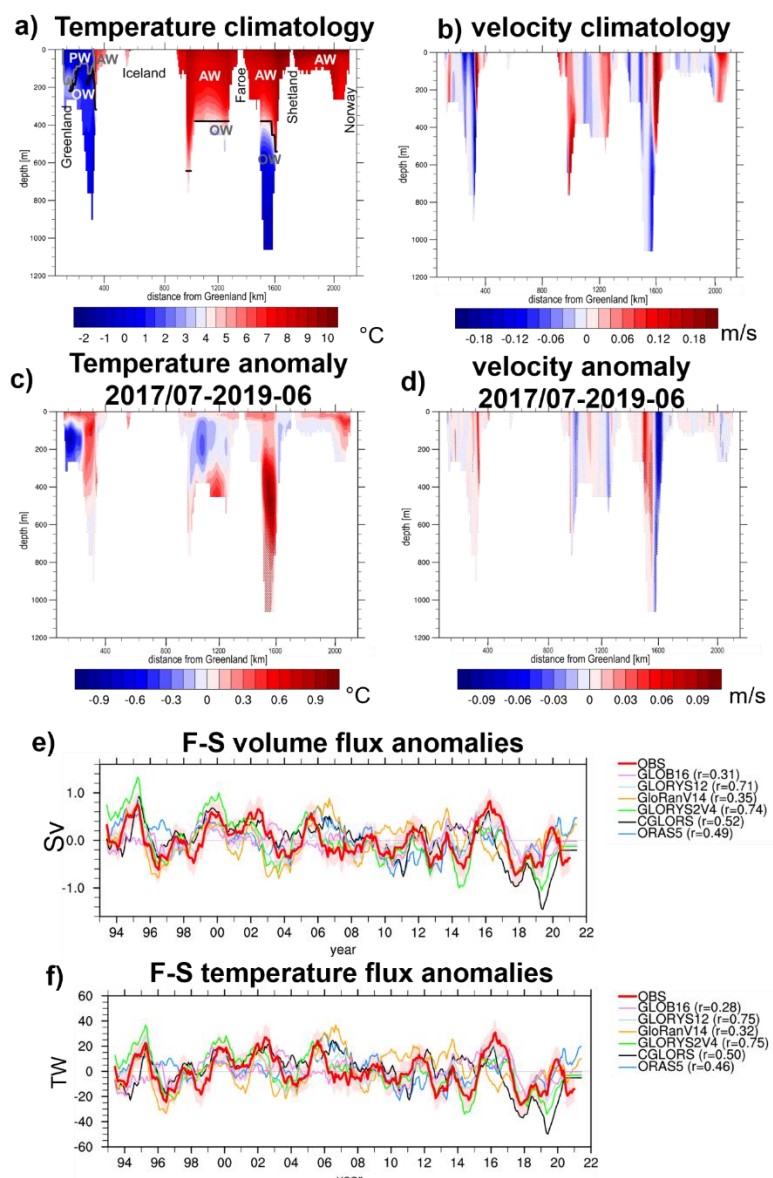


**Fig. 2.** Longitude-depth sections of a) mean temperature (with water mass boundaries indicated), b) mean velocity, c) 2017/07-2019/06 anomalous temperature, and e) 2017/07-2019/06 anomalous velocity across GSR based on GLORYS12 (stippling denotes grid cells where anomalies are >2σ of biennial anomalies). Note that the section does not everywhere go along the shallowest part

of Denmark Strait and I-F channel, leading to deeper trenches in some places; time series of e) anomalous volume and f) anomalous temperature flux through Faroe-Shetland Branch, where the red shading indicates ± 1 standard error of the observational data. Temporal correlations of reanalyses with observations are provided in the legends.




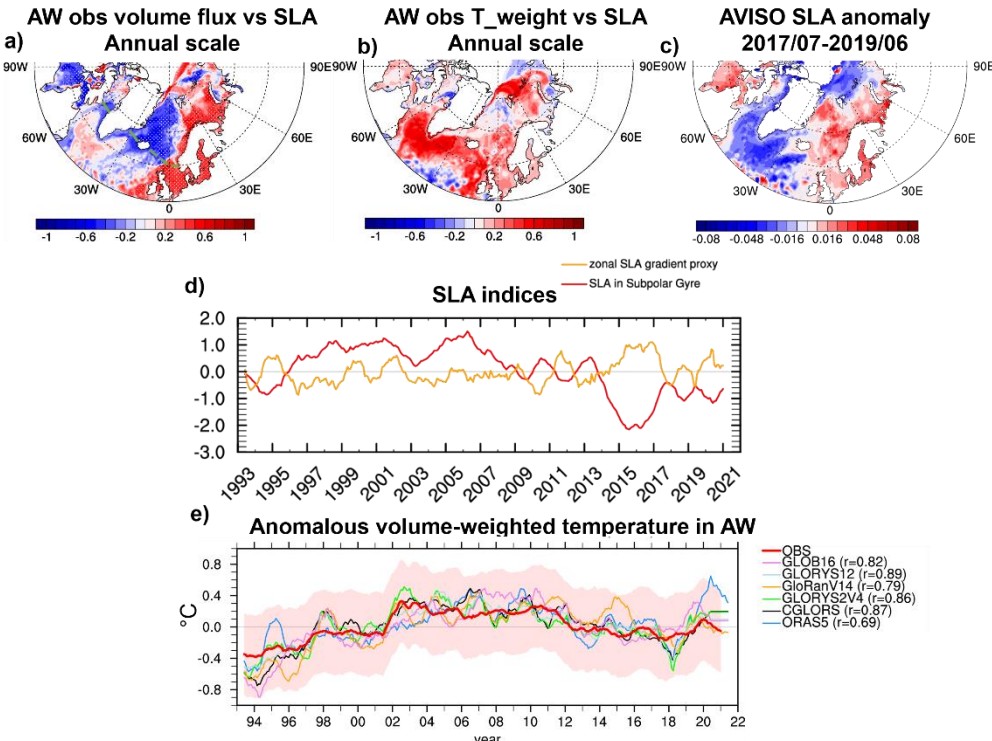

**Fig. 3.** Correlation of observation-based a) AW volume flux and b) AW flux-weighted temperature with SLA with a 12-monthly smoother applied; c) SLA anomaly in 2017-07-2019-06; d) temporal evolution (standardized) of two SLA-based indices based on the zonal SLA gradient at GSR and the SLA in the SPG region; e) temporal evolution of volume-weighted temperature anomalies of Atlantic waters at the GSR (red shading indicates ± 1 standard error of the observational data and temporal correlations of reanalyses with observations are provided in the legend); a) additionally shows the location of Davis Strait and GSR sections in green. Stippling in a) and b) denotes statistically significant correlations on the 95% confidence level.
