# Peer review of "Recent variations in oceanic transports across the Greenland-Scotland-Ridge"

_State of the Planet, 2022_

## Author Comment (AC1)

*Response to Reviewer #1*

*Understanding ongoing Arctic climate change is high on the international agenda, and the variability of ocean heat transport across the Greenland-Scotland-Ridge is important, as it is the main exchange between the Arctic (Nordic Seas) and the Atlantic. I find many aspects of the paper correct, and some of the conclusions drawn by the authors are well funded in my view. There is however one major issue that needs to be sorted, and there should be plenty of room for improvements in such a short paper.*

*The paper is a decent attempt at comparison of presently available ORA's (Ocean Re-Analysis) and observations. The paper confirms three basic expectations and there are relevant citations given in these cases; a) AW volume inflow dominates GSR heat transport, and b) there is a larger seasonal volume/heat transport during winter, c) the AW heat transport dominates heat transport to the Arctic. There is also one new important change documented; that the large ocean-heat-transport pulse found by Tsubouchi et al (2021) for 2003-2012 has not continued.*

We are glad our main points have come across and would like to thank the reviewer for providing constructive comments. Please see below for our responses in black. Before this, we would like to point out two changes that we plan to include in the revised manuscript:
1) We recently discovered an error in the computation of ORA-based transports. Correction of the error leads to generally improved correlations with observation. Correction of the error also leads to an increase of the mean Atlantic Water (AW) volume flux diagnosed from the lower resolution (GREP) products, i.e. their low bias in *total* AW influx is reduced. Nevertheless, the conclusions about higher resolution ORAs exhibiting a better *distribution* of inflow across the different branches still holds. We will revise the manuscript to reflect these changes.
2) Reviewer #2 criticized that differences between GLOB16 (the forced ocean run at 1/16 degree resolution without data assimilation) and lower-resolution ORAs were not necessarily attributable to resolution as we did not include a low-resolution control run for a clean comparison. To address this, we now include results from GLOB4 (a ¼ degree version of GLOB16). Yet, we will limit the discussion to their representation of volume fluxes (i.e. ocean circulation) to strengthen the conclusions about resolution-dependence of circulation in the ORAs. Heat fluxes from GLOB4 and GLOB16 will not be discussed as they exhibit temperature (and hence heat flux) biases (due to their nature of being non-assimilating runs) that detract from the main points of the study.

*The main problem with the paper that it is not comparing apples and apples.*

The reviewer is right that the comparison between observations and reanalyses is not straight-forward, but, as laid out below, we are convinced our methods are appropriate to allow for a fair comparison. We realize this important point is hardly discussed in the submitted manuscript. We will improve on this by adding discussion of this issue in the appropriate places (see below).

*The ORA based values are (supposedly) net volume flux values calculated on the native grid and at the shortest distance between for example Iceland and the Faroes (Shown in Fig 3 a, but too small, and should be shown clearly in an early figure).*

We will include a new figure showing clearly the location of the moorings (in yellow) going into the observational estimate and the ORA-based section (green line), as shown below.

[Figure]

*This is not what is observed. The observations are drawn from a section running north from the Faroe Island (an OK map is found in Østerhus et al 2019 (Fig. 3)). I suspect that there is a large re-circulation of AW across the sill that is not observed. The observations document the flow north of the Faroes, and some of this water is likeley to cross the GSR again between the Faroes and Shetland. So the ORA based values are NET inflow, but the observations have an un-known portion of water that re-circulates and leaves shortly after entering the GSR.*

In the paper, we use the long-term observations of the exchanges across the Greenland-Scotland Ridge that were initiated in the Nordic WOCE project in the mid 1990ies. Both for the North Icelandic Irminger Current (NIIC) and the Iceland-Faroe Branch, the most appropriate location to monitor these inflows was considered to be just downstream of the Ridge, where these branches are focused into narrow currents. The observations, therefore, exclude recirculation that might happen within the Denmark Strait and on the Iceland-Faroe Ridge, which are also excluded in the ORA-based net fluxes. The effect of further recirculation between the Ridge crest and the monitoring sites is considered minor. Within the Faroe-Shetland Channel (FSC), retroflections of the Faroe Current is known to occur. The monitoring section was selected to run along a historical hydrographic standard section (Berx et al, 2013) and both the inflowing Continental Slope current on the Scottish side and the re-circulated South Faroe Current on the Faroese side have been monitored for several years with ADCPs (Berx et. al. 2013). The transport time series for the FSC therefore represents net volume flux of AW excluding circulation on the shelfs (updated from Berx et al, 2013). The question if any Atlantic Water returns back across the GSR has also been studied. Using hydrographic observations, current measurement, Satellite Sea Level observations and surface drifters, Hansen et. al. (2017) do not find evidence for a continuous flow of Atlantic water from the Faroe–Shetland Channel to the Faroe Bank Channel over the Faroese slope as indicated in Rossby and Flagg (2012). On the other hand, a clockwise circulation of shelf water is on the Faroe Shelf (Larsen et al, 2008), but, as mentioned above, it is not included in the observations of the Iceland-Faroe nor the Faroe-Shetland AW transports. Based on the results in Hansen et al (2017) and the improved flux estimates in Rossby et al (2018) we do not find that a recirculation of AW back across the GSR is of major importance.

Overall, the uncertainty of the AW inflow observations at Hornbanki are estimated to be <15%, as described in Jónsson and Valdimarsson (2012). At the Iceland-Faroe branch, the uncertainty is also <15% (Hansen et al, 2015), while the uncertainty of the Faroe-Shetland branch is slightly higher (<19%) probably due to the retroflection that occurs in the channel (Berx et al, 2013).

Based on this and our response further above, we will insert the following paragraph to the data and method section:

**"We note that quantification methods of oceanic transports in reanalyses and observations are fundamentally different, which needs to be kept in mind when intercomparing. While the former estimate is based on surface to bottom, coast to coast temperature and velocity sections across the Arctic Mediterranean, the latter estimate is based on the sum of 11 major ocean current transport estimates that is categorized into three major water masses – AW, PW and OW (Tsubouchi et al., 2021). An assumption is that the 11 major ocean currents represent well the major water mass exchanges across the Arctic Mediterranean. This means it is important that no recirculation, e.g. of AW waters, remains unobserved, as this would introduce biases to the observational estimate. This assumption has been assessed and confirmed many times over the last two decades from establishment of sustained hydrographic sections in GSR in the 1990ies (e,g, Dickson et al., 2008) to recent oceanographic surveys to capture ocean circulation in GSR for AW (e.g. Berx et al., 2013, Hansen et al., 2017, Jónsson and Valdimarsson, 2012) and OW (Hansen et al., 2018). We also note that remaining uncertainties arising from potential undersampling are taken into account in the observational estimate obtained through the inverse model."**

*There is actually one substantial paper that shows observations (and re-circulations) between the islands that is not cited; Rossby et al (2018). This is quite strange. In special as the net flow in Rossby et al (2018) is close to zero across the GSR. This is a substantial difference that needs to be explained, and one would think that the (fairly) high resolution ORA's could provide an answer here. The net northward GSR volume transport found by Mayer et al of +0.7 Sv is comparable to what is found by others though. For example Smedsrud et al (2022) found +1.0 Sv as the centennial mean, and some net northward GSR flow must exist to Balance the net southward flow west of Greenland of about -1.7 Sv (Tsubouchi et al 2021). Bering Strait inflow is only about +0.8 Sv.*

For the observational estimate, net zero transport around the Arctic Mediterranean is achieved by the inverse model every single month. On average, net northward GSR flow (+0.7 Sv) is balanced by net southward flow west of Greenland (-1.7 Sv), Bering Strait inflow (+0.8Sv) and surface freshwater input (+0.2 Sv).
Ocean reanalyses conserve volume and those with data assimilation (i.e. all but GLOB16) obtain a similar volume budget across GSR, Davis Strait, Bering Strait as the observational estimate (average volume fluxes from GREP are 1.2±0.6 Sv, -2.6±0.6 Sv, 1.3±0.1 Sv, respectively).

*While the observations around the (Iceland, Faroes, Shetland) islands are subjected to fundamental problems regarding re-circulation, the ones along the Svinøy Section are not (Orvik 2022). Here the AW is well north of the GSR, and this would infact be a much better location for direct comparison. The Svinøy data appears consistent though; there are no trends in Volume or Temperature over the period 1995-2020, similar to Figure 1 e) f).*

As explained above, Hansen et al (2017) have shown, that recirculation in the Faroe Shetland Channel is not a fundamental problem, and hence the chosen section is deemed appropriate. Moreover, Orvik (2022) only covers the western Norwegian Atlantic Current and not the eastern NwAC branch, which is covered in the observations of the Iceland Faroe inflow used here. Another point is the fact that a direct comparison of the ORA data with observations at

Svinøy Section does not guarantee an "apples and apples" comparison, as the exact location of AW major flow path in the ORAs may differ from that in observations.

*The re-circulation of AW appears on my side to be the major explanation for the difference between the ORA's and the observations. For example, there are no observations of southward AW flow east of Shetland. This was recently noted by Smedsrud et al (2022) where the simulated AW inflow mean (1900-2000) was as high as +9.5 Sv, but there is also a net AW outflow of -3.3 Sv. They had some hydrographic and current meter data to compare with, although it is a bit hard to understand those calculations.*

When comparing ORAs with observations (or different observational methods and even when comparing different ORAs) we cannot expect a perfect match. Actually, taking uncertainties into account, GLORYS12 overall compares very well with the observations. The reviewer highlights the +9.5 Sv AW inflow by Smedsrud et al (2022) and points to their AW outflow (recirculation) of -3.3 Sv, but the reviewer does not mention their very low (-3.3 Sv) Overflow across the GSR, which is neither in agreement with the GSR observations nor the ORAs. Moreover, Smedsrud et al (2022) provide a centennial estimate (1900 – 2000), which only partly overlaps with our shorter period.

*The 'high bias' in volume transport from observations would carry onto the ocean heat transport.*

As the reviewer says, it *would*: but as explained in detail above, we are convinced our observation-based volume flux estimates do not exhibit a high bias (in contrary to the reviewer's suspicion), and hence we do not expect our observation-based heat transport estimate to be unduly high.

*The Mayer et al (2022) estimate of a total surface heat loss of about 350 TW does not appear realistic. There are fundamental problems with atmospheric stratification and atmospheric boundary-layer parametrizations that need to be examined before such a high value can be accepted. If the ocean cannot transport this heat, it cannot leave the ocean surface. And the Barents Sea and the Arctic Ocean is the most rapidly warming ocean on the planet, so heat is accumulating too…*

The energy-budget-based estimate of oceanic heat transport presented by Mayer et al. (2022a) and cited in our manuscript combines an estimate of net surface energy flux inferred from the atmospheric energy budget and takes into account oceanic heat storage and sea ice melt. This approach is well-established and errors are deemed moderate on a regional scale (see Mayer et al 2022b for a detailed assessment). For example, Trenberth and Fasullo (2017), Liu et al. (2020), or Mayer et al. (2022b) successfully used this method to infer ocean heat transports at the RAPID section, with remarkably good agreement with observations (both in terms of means and variability), which builds confidence to use this approach also for inferring oceanic heat transports into the Arctic Mediterranean.

We emphasize that the employed fields of net surface energy flux are *not* the parameterized surface fluxes obtained from short-term forecasts of a reanalysis. We agree that those fields are oftentimes biased due to uncertainties in the quantities going into the parameterizations, uncertainties in the parameterizations themselves, and the reduced observational constraint (since the surface flux fields from reanalyses are typically based on short-term forecasts rather than analyses). Rather, the net surface fluxes estimate combines net top-of-the-atmosphere fluxes as measured by satellites (CERES-EBAF, the Arctic Mediterranean regional bias of which is deemed very small: ~7TW; see Mayer et al. 2019) and the divergence of atmospheric

lateral energy transports as diagnosed from ERA5 analyses, which is much better constrained by observations than parameterized fluxes, as has been discussed many times (see, e.g., von Schuckmann et al. 2016). Certainly, the divergence of atmospheric energy transports exhibits uncertainties, although constrained by all kinds of observations. Mayer et al. (2022a) found an Arctic Mediterranean regional mean difference of 27.5 TW between the divergence from two different reanalysis products ERA5 and JRA55 (the latter actually suggesting even higher transports), which is small in a relative sense (only ~2.5% of the mean convergence of atmospheric transports order 1.1 PW).

We reiterate that the inferred estimate of ocean heat transport also takes into account ocean heat storage (in the form of warming and sea ice melt; order 13 TW) and the energetic effect of sea ice transports (see Mayer et al. 2022a, but also Mayer et al. 2021).

Based on the above explanations, we do not believe that the indirect heat transport estimate is missing any major terms, but of course an indirect estimate always suffers from accumulation of errors in other terms.

In conclusion, we did not mean to postulate that our estimate of 348 TW is a more realistic value than the observation-based estimate of 302 TW, but it indicates that our observation-based estimate is unlikely to be biased high and the ORAs are very likely biased low. **This will be phrased more carefully in the revised manuscript.**

*The main point is that the observations and ORA's show different things, and you cannot compare them directly before you extract the flow where it is measured as well. With the increased resolution of the ORA's it should now be possible to directly compare the observations with the simulated flow exactly where it is measured. Possibly this discrepancy can be solved, but until then I suspect that the ORA's are just as correct as the «observations» and that the true OHT value is closer to 200 TW than to the «observed» 300 TW.*

As explained above, there are several arguments why the observation-based heat flux estimate of 302 TW is be deemed realistic: i) AW recirculation as a reason for a high AW volume flux bias likely plays a minor role (see responses above as well as provided references), ii) the higher resolution ORA GLORYS12, with an undoubtfully more realistic representation of the flow across the GSR than the ¼° products, exhibits volume and heat flux in closer agreement with observations than the lower resolution product, and iii) the energy-budget-based estimate of oceanic heat flux of 348 TW does not support the assumption that the observational estimate is biased high. **These points will be made more clearly in the revised manuscript.**

*The paper was otherwise nicely written, and I have no detailed list of corrections. There is a proper map missing showing where the current meter and hydrographic sections are, and likewise where the grid cells for the ORA-based estimates are located.*

We will include a map as suggested, please see above.

*Most of the work done is fine, you just need to do a proper comparison, and discuss the other possibility that the ORA's are more correct than you think.*

*Abstract: This is wrong: "The considered ocean reanalyses underestimate the observed 1993-2021 mean inflow of warm and saline Atlantic Waters of 8.0 ± 0.5 Sv by 7-23%". What you have shown so far is that the simulated NET is lower than the observed inflow. There is an (unknown) volume of observed outflow, so the simulated NET could well be perfectly correct.*

We will clarify by adding "net" before "warm and saline Atlantic Waters".

*Conclusion: This also needs to be modified; "The energy-budget based estimate from Mayer et al. (2022) suggests even higher net heat flux than oceanic observations, which confirms the underestimation of transports by the ocean reanalyses and indicates that observations possibly miss some influx of heat."*

*It is good to have confidence in one's own results, but as a scientists we must also be open for that they are wrong. In this case the observations are likely not showing what you state they do. They show inflow values, and does not give a proper estimate of the net flow.*

As explained above, the energy-budget-based estimate is not meant to represent a more realistic estimate than observations, but adds evidence that the estimate of 300 TW net heat transport is unlikely to be biased high. We agree however that the stated implication of the observational estimate being biased low may be too strong. **We will tone it down in the revised manuscript.**

*Suggested new references:*

*Orvik, K. A. (2022). Long-term moored current and temperature measurements of the Atlantic inflow into the Nordic Seas in the Norwegian Atlantic Current; 1995–2020. Geophysical Research Letters, 49, e2021GL096427. https://doi.org/10.1029/2021GL096427*

*Rossby, T., Flagg, C., Chafik, L., Harden, B., & Søiland, H. (2018). A direct estimate of volume, heat, and freshwater exchange across the Greenland-Iceland-Faroe-Scotland Ridge. Journal of Geophysical Research: Oceans, 123, 7139–7153. https://doi.org/10.1029/2018JC014250*

*Smedsrud et al (2022) Nordic Seas heat loss, Atlantic inflow, and Arctic sea ice cover over the last century. Rev. Geophysics.*

Thanks for these new references. Rossby et al (2018) provide flux estimates for the whole GSR and will be added to the revised manuscript. The other references do not quite overlap spatially (Orvik, 2022) or temporally (Smedsrud et al, 2022), but we will consider if they can be added to the revised manuscript.

New Refs in the reply:

Childers, K. H., C. N. Flagg, and T. Rossby (2014), Direct velocity observations of volume flux between
Iceland and the Shetland Islands, J. Geophys. Res. Oceans, 119, 5934–5944, doi:10.1002/2014JC009946.

Hansen, B., Poulsen, T., Húsgarð Larsen, K. M., Hátún, H., Østerhus, S., Darelius, E., Berx, B., Quadfasel, D., and Jochumsen, K.: Atlantic water flow through the Faroese Channels, Ocean Sci., 13, 873–888, https://doi.org/10.5194/os-13-873-2017, 2017.

Hansen, B., Larsen, K. M. H., Olsen, S. M., Quadfasel, D., Jochumsen, K., and Østerhus, S.: Overflow of cold water across the Iceland–Faroe Ridge through the Western Valley, Ocean Sci., 14, 871–885, https://doi.org/10.5194/os-14-871-2018, 2018.

Jónsson, S., and Valdimarsson, H. 2012. Water mass transport variability to the North Icelandic shelf, 1994–2010. – ICES Journal of Marine Science, 69: 809–815. https://doi.org/10.1093/icesjms/fss024

Larsen, K. M. H., Hansen, B., Svendsen, H. 2008. Faroe Shelf Water. Continental Shelf Research, 28 (14) https://doi.org/10.1016/j.csr.2008.04.006.

Rossby, T and Flagg, C. N., 2012. Direct measurement of volume flux in the Faroe-Shetland Channel
and over the Iceland-Faroe Ridge. GEOPHYSICAL RESEARCH LETTERS, VOL. 39, L07602, doi:10.1029/2012GL051269, 2012

Rossby, T., M. D. Prater, and H. Søiland (2009), Pathways of inflow and dispersion of warm waters in the Nordic seas,
J. Geophys. Res., 114, C04011, doi:10.1029/2008JC005073.

Trenberth, K. E., & Fasullo, J. T. (2017). Atlantic meridional heat transports computed from balancing Earth's energy locally. Geophysical Research Letters, 44(4), 1919-1927.

Liu, C., Allan, R. P., Mayer, M., Hyder, P., Desbruyères, D., Cheng, L., ... & Zhang, Y. (2020). Variability in the global energy budget and transports 1985–2017. Climate Dynamics, 55(11), 3381-3396.

Mayer, M., Tietsche, S., Haimberger, L., Tsubouchi, T., Mayer, J., & Zuo, H. (2019). An improved estimate of the coupled Arctic energy budget. Journal of Climate, 32(22), 7915-7934.

Mayer, M., V.S. Lien, K.A. Mork, K. von Schuckmann, M. Monier, E. Greiner (2021): Ocean heat content in the High North, in CMEMS Ocean State Report Vol 5, Journal of Operational Oceanography 14:sup1, 1-185, DOI: 10.1080/1755876X.2021.1946240.

Mayer, M., Tsubouchi, T., von Schuckmann, K., Seitner, V., Winkelbauer, S., Haimberger, L., (2022a). Atmospheric and oceanic contributions to observed Nordic Seas and Arctic Ocean Heat Content variations 1993-2020. In: Copernicus Ocean State Report, Issue 6, Journal of Operational Oceanography, 15:sup1, s119–s126; DOI: 10.1080/1755876X.2022.2095169

Mayer, J., Mayer, M., & Haimberger, L. (2021). Consistency and homogeneity of atmospheric energy, moisture, and mass budgets in ERA5. Journal of Climate, 34(10), 3955-3974.

Mayer, J., Mayer, M., Haimberger, L., & Liu, C. (2022b). Comparison of Surface Energy Fluxes from Global to Local Scale, Journal of Climate, 35(14), 4551-4569.

von Schuckmann, K., and Coauthors, 2016a: Report of the 1st workshop of CLIVAR research focus CONCEPT-HEAT. WCRP Rep. 6/2016, 23 pp.,
http://www.clivar.org/documents/report-1st-workshop-clivar-research-focus-concept-heat.

---

## Author Comment (AC2)

*This short paper presents an inter-comparison of transports in correspondence with the GSR section from several state-of-the-art reanalyses and simulations. It also includes a more detailed analysis of the recent decrease in heat fluxes after 2017, with some explanations and a proposal for sea level-based indexes. As such, it is an interesting piece of work, useful for the oceanic community, and as a reference for future validation and process-oriented studies. I recommend publishing this work after the authors address a few general comments.*

Thanks for the appreciative comments. Please see below for our responses in black. Before this, we would like to point out two changes that we plan to include in the revised manuscript: 1) We recently discovered an error in the computation of ORA-based transports. Correction of the error leads to generally improved correlations with observation. Correction of the error also leads to an increase of the mean Atlantic Water (AW) volume flux diagnosed from the lower resolution (GREP) products, i.e. their low bias in *total* AW influx is reduced. Nevertheless, the conclusions about higher resolution ORAs exhibiting a better *distribution* of inflow across the different branches still holds. We will revise the manuscript to reflect these changes.

2) Reviewer #2 criticized that differences between GLOB16 (the forced ocean run at 1/16 degree resolution without data assimilation) and lower-resolution ORAs were not necessarily attributable to resolution as we did not include a low-resolution control run for a clean comparison. To address this, we now include results from GLOB4 (a ¼ degree version of GLOB16). Yet, we will limit the discussion to their representation of volume fluxes (i.e. ocean circulation) to strengthen the conclusions about resolution-dependence of circulation in the ORAs. Heat fluxes from GLOB4 and GLOB16 will not be discussed as they exhibit temperature (and hence heat flux) biases (due to their nature of being non-assimilating runs) that detract from the main points of the study.

*1) Table 2 and the figures are not consistent: the figures report all the GREP members while the Table only the GREP ensemble mean. I recommend the authors report the values of all GREP members and their ensemble mean. Furthermore, the table should contain more diagnostics and include those already presented in the text (temporal standard deviation, correlation values, etc.). This could be very useful for future quantitative studies.*

We agree a more detailed display of metrics in table 2 could be useful as a reference for future studies. However, this expansion needs to be balanced with the space constraints for OSR contributions. We thus suggest to expand the table to display the averages for all GREP reanalyses separately, not only the GREP ensemble mean. We note that the correlation coefficients are provided in the time series plots and thus do not necessarily have to be replicated in the table.

*2) The inclusion of GLOB16 is not sound. This is a free-running model experiment, at a higher resolution than the other products, and a corresponding assimilation experiment does not exist. In practice, its inclusion poses more questions and does not help to answer any question. Using an ensemble of reanalyses helps to identify the envelope of uncertainty; using GLORYS12 helps address the impact of spatial resolution. But it is not clear whether results from GLOB16 can be ascribed to different resolutions? different model configuration? lack of observational constraints? different initial conditions? I miss the point for including it.*

To address this point, we  assessed the ¼ deg counterpart of GLOB16, named GLOB4, which allows for a clear attribution of differences to resolution. Indeed, the higher resolution run exhibits a distribution of volume fluxes across the different branches crossing the GSR that are in better agreement with observations compared to GLOB4. We propose to only discuss

these runs in terms of volume fluxes / circulation, i.e. to strengthen the points regarding resolution of the ORAs. Heat transports from GLOB16 and GLOB4 will not be discussed, as their temperature biases lead to biased heat flux estimates.

*3) Section 3.3 is not much conclusive either. One suggestion from the authors is that SPG strength induces a delayed response in the AW temperature. However, lagged correlations and similar tools could be used to provide a quantitative answer about the fitness of the SLA-based indexes for capturing the AW inflow variability.*

Good suggestion. Cross-correlation analysis of AW temperature and the SPG strength seems to support our statement by showing maximum positive correlation when AW temperature lags by ~2-3 years, with maximum correlation coefficients between 0.3 and 0.5 (depending on the considered data set). However, these correlations are not statistically significant because of the high auto-correlation of the involved time series, which reduces the effective degrees of freedom. This result will be mentioned in the revised manuscript.

*Minor point*

*In line 46: I suggest referring to "the so-called Arctic Mediterranean" as it might be non-obvious for a general readership*

Agreed.

---

## Author Response (AR2)

Following referee#2'2 comment, we added two additional statements about the differences how transports from the reanalyses and observations are derived (see below). We additionally changed two words in the last paragraph to avoid word repetition and made one small clarification in the legend of Fig. 4.

In section 2, we changed the paragraph

*We note that quantification methods of oceanic transports in reanalyses and observations are fundamentally different, which needs to be kept in mind when intercomparing. While the former estimate is based on surface to bottom, coast to coast temperature and velocity sections across the Arctic Mediterranean, the latter estimate is based on the sum of 11 major ocean current transport estimates that is categorized into three major water masses – AW, PW and OW (Tsubouchi et al., 2021).*

to

*We note that quantification methods of oceanic transports in reanalyses and observations are fundamentally different, which needs to be kept in mind when intercomparing. The reanalysis-based estimate is based on surface to bottom, coast to coast temperature and velocity sections across the Arctic Mediterranean. This ensures conservation of volume and avoids projection of potentially biased positioning of currents in the reanalyses onto the transport estimates. The observational estimate is based on the sum of 11 major ocean current transport estimates that is categorized into three major water masses – AW, PW and OW (Tsubouchi et al., 2021).*

In section 5 (conclusions), we changed

*Reanalysis-based oceanic transports show generally good agreement with observations on the scale of single branches of the GSR, both in terms of mean and variability of volume and heat fluxes. There is some indication that the higher resolution products have a better representation of AW inflow in the I-F and F-S branches. All considered products underestimate net heat flux into the Arctic Mediterranean. The magnitude of the low bias is correlated with the strength of AW volume flux but a warm bias in OW and cold bias in Davis Strait inflow further add to the found net heat flux bias. The energy-budget-based estimate from Mayer et al. (2022a)…*

to

*Reanalysis-based oceanic transports show generally good agreement with observations on the scale of single branches of the GSR, both in terms of mean and variability of volume and heat fluxes. There is some indication that the higher resolution products have a better representation of AW inflow in the I-F and F-S branches. All considered products underestimate net heat flux into the Arctic*

*Mediterranean. The magnitude of the low bias is correlated with the strength of AW volume flux but a warm bias in OW and cold bias in Davis Strait inflow further add to the found net heat flux bias. We reiterate that reanalysis-based and observational transport estimates are obtained in different ways (closed line integrations versus measurements from 11 branches with an inverse model applied) but, as elaborated in section 2, we deem this a fair and robust approach for an intercomparison. The energy-budget-based estimate from Mayer et al. (2022a)…*